# Influence of Simulated State of Disc Degeneration and Axial Stiffness of Coupler in a Hybrid Performance Stabilisation System on the Biomechanics of a Spine Segment Model

**DOI:** 10.3390/bioengineering10091042

**Published:** 2023-09-05

**Authors:** Chih-Kun Hsiao, Hao-Yuan Hsiao, Yi-Jung Tsai, Chao-Ming Hsu, Yuan-Kun Tu

**Affiliations:** 1Department of Medical Research, E-Da Hospital, I-Shou University, Kaohsiung 824, Taiwan; shiaujk@gmail.com (C.-K.H.); ed108805@edah.org.tw (Y.-J.T.); 2Department of Orthopedics, E-Da Hospital, I-Shou University, Kaohsiung 824, Taiwan; m8571409@yahoo.com.tw; 3Institute of Medical Science and Technology, National Sun Yat-sen University, Kaohsiung 804, Taiwan; 4Department of Mechanical Engineering, National Kaohsiung University of Science and Technology, Kaohsiung 824, Taiwan

**Keywords:** hybrid performance system, disc degeneration, adjacent segment, FE model

## Abstract

Spinal fusion surgery leads to the restriction of mobility in the vertebral segments postoperatively, thereby causing stress to rise at the adjacent levels, resulting in early degeneration and a high risk of adjacent vertebral fractures. Thus, to address this issue, non-fusion surgery applies some pedicle screw-based dynamic stabilisation systems to provide stability and micromotion, thereby reducing stress in the fusion segments. Among these systems, the hybrid performance stabilisation system (HPSS) combines a rigid rod, transfer screw, and coupler design to offer a semi-rigid fixation method that preserves some mobility near the fusion site and reduces the adjacent segment compensatory effects. However, further research and confirmation are needed regarding the biomechanical effects of the dynamic coupler stiffness of the HPSS on the intrinsic degenerated adjacent segment. Therefore, this study utilised the finite element method to investigate the impact of the coupler stiffness of the HPSS on the mobility of the lumbar vertebral segments and the stress distribution in the intervertebral discs under flexion, extension, and lateral bending, as well as the clinical applicability of the HPSS on the discs with intrinsic moderate and severe degeneration at the adjacent level. The analytical results indicated that, regardless of the degree of disc degeneration, the use of a dynamic coupler stiffness of 57 N/mm in the HPSS may reduce the stress concentrations at the adjacent levels. However, for severely degenerated discs, the postoperative stress on the adjacent segments with the HPSS was still higher compared with that of the discs with moderate degeneration. We conclude that, when the discs had moderate degeneration, increasing the coupler stiffness led to a decrease in disc mobility. In the case of severe disc degeneration, the effect on disc mobility by coupler stiffness was less pronounced. Increasing the coupler stiffness ked to higher stress on intervertebral discs with moderate degeneration, while its effect on stress was less pronounced for discs with severe degeneration. It is recommended that patients with severe degeneration who undergo spinal dynamic stabilisation should remain mindful of the risk of accelerated adjacent segment degeneration.

## 1. Introduction

Spinal fusion surgery is a surgical technique commonly employed for treating intervertebral disc degeneration, spinal stenosis, and vertebral slippage. The adaptation of the posterior approach for this surgery uses the insertion of pedicle screws into the vertebrae from the posterior side, and then metal rods to connect them. In some cases, interbody fusion cages are also used to achieve spinal fusion. The primary purpose of this fusion surgery is to achieve spinal stability, especially in cases where the spine is unstable. However, despite advances in fusion surgery techniques, higher fusion rates do not always result in satisfactory clinical outcomes [1,2]. Clinical observations have revealed that some patients experience early or accelerated degeneration [3], or adjacent segment fractures following fusion surgery [4,5]. Biomechanical studies have identified that fusion surgery can result in a restricted range of motion (ROM) in the adjacent segments, which can lead to either a rise in stress or the stress concentration phenomenon [6,7]. This phenomenon is a significant cause of early adjacent segment degeneration or adjacent segment fractures [8,9]. Previous studies suggest that the accelerated adjacent segment disc degeneration following surgery is not solely due to the reinforcement effect of the fused or instrumented segment, but can also be influenced by changes in the stress distribution in the adjacent segments after fusion surgery [10,11]. Therefore, although spinal fusion surgery and the use of spinal internal fixation devices can increase spinal stability, they can also alter the biomechanical characteristics of the spine [12,13]. While these procedures have the advantage of reducing pain and restoring the lost functions in the affected area, they can also lead to complications in the adjacent segments [14,15].

Several posterior non-fusion surgical systems have been developed to reduce the risk of adjacent segment pathology caused by fusion surgery. The non-fusion dynamic stabilisation system (DSS) is one of the new fixation devices. It provides flexible or semi-rigid fixation, which aims to preserve some degree of motion in the fusion area, reduce the rigidity of the fusion segment, and minimise the stress concentration effects caused by any adjacent segment compensation. The goal of this device is to prevent early degeneration in the adjacent segment caused by the fused segment. Various dynamic stabilisation systems based on this concept have been used clinically and provided favourable outcomes. Among DSSs, the DSS^®^ stabilisation system is primarily used as a standalone for lumbar pedicle fixation and is suitable for the T4 to S1 vertebrae, where it provides stability in the treatment of thoracic, lumbar, and sacral deformities [13,16]. This system consists of polyaxial screws, pedicle screws, slotted couplers, and rigid couplers, which allow for various fixation configurations that match the lumbar anatomy of the patients. Its main indications include intervertebral disc degeneration or pathology, degenerative scoliosis, lumbar spinal stenosis, or vertebral slippage. Since the introduction of the DSS^®^, a hybrid performance stabilisation system (HPSS) was developed. The system consists of a fusion level and a dynamic level, with the fusion level utilising steel rods for multilevel fusion and being connected to the dynamic segment with transitional screws (Figure 1) [17]. The coupler is designed for the dynamic level to preserve slight mobility, reduce compensation effects caused by fusion, and minimise the risks of postoperative pathology and complications resulting from the high stress generated by the fusion. This system has been increasingly adopted in clinical practices.

One key design characteristic of the HPSS is the stiffness of the dynamic coupler [18]. Although the coupler in the adjacent segment has a flexible stiffness to reduce stress and alter the motion of the vertebra, clinical evidence has shown that the adjacent segment often exhibits some degree of degeneration [19]. Thus, when the adjacent vertebrae exhibit different levels of degeneration, it is necessary to consider the design stiffness of the coupler and its compensatory effects on the intervertebral disc stress and segmental motion in both the dynamic and adjacent levels [20,21]. Currently, biomechanical studies that address this aspect are lacking. Thus, the purpose of the present study was to use the finite element analysis (FEA) method to determine the influence of the simulated state of degeneration of two intervertebral discs (IVDs) (at L3/L4 and L4/L5) in a model of a thoracolumbar spine section (T12–S1) and the axial stiffness (K) of the dynamic coupler in the HPSS on changes in two biomechanical responses of the model under clinically relevant loadings. The responses were the mean von Mises stress at each of the IVDs and the range of motion (ROM) at each of the six segments. The changes were with respect to the corresponding values in a model of a healthy spine section.

## 2. Methods

### 2.1. FE Model Construction

To understand the stress in the adjacent intervertebral discs during spinal fusion with flexion, a three-dimensional (3D) finite element (FE) model of the human spine was constructed from a cadaveric spine (62-year-old male) using computed tomography (CT) images (DICOM format) of T12–S1 taken from 64 slices (0.625 mm intervals). The CT images were imported into MIMICS software (Materialise Mimics Innovation Suite Medical 16.0) to establish the 3D geometric structure (solid model) of the T12–S1 segments in STL format. SolidWorks (Dassault Systems 2018, SolidWorks Corp., Waltham, MA, USA) was used to perform the smoothing process to remove the spikes and holes on the vertebral surface and construct the intervertebral disc. Then, the geometric model was imported into ANSYS software (ANSYS Ltd., Canonsburg, PA, USA) to mesh and generate the finite element (FE) model (Figure 2).

The FE model comprised the intact and implanted models. In the intact finite element model, the bony structures of the vertebral segment included were the cortical bone and the cancellous bone. The thickness of cortical bone was set to 1.5 mm. The endplates were constructed as a shell with a thickness of 0.5 mm. The facet joints were assumed to have a 0.5 mm thick cartilage with surface-to-surface, frictionless sliding contact. Eight-noded solid elements were used to model each of these bones. Each bone was taken to be a linear elastic and isotropic material. Both the L3/L4 and L4/L5 discs were assigned the same degree or grade of degeneration, while the other levels were healthy. The disc in the FE model included normal (healthy), moderate, and severe degeneration. The intervertebral disc consisted of annulus fibre layers and nucleus pulposus. Tetrahedral meshing was performed for all intervertebral discs and screw–rod systems. The ligaments in the FE model included the anterior and posterior longitudinal ligaments and the supraspinal ligaments. The tension-only spring elements were used to model all ligaments. Linear and isotropic material properties for cortical bone, cancellous bone, posterior bony elements, annulus fibre layers, and nucleus pulposus were employed in this study. Studies have shown that degenerated discs are more uniform than healthy discs, and an “effective modulus” can be used to characterise the complex behaviour of a real disc [22]. Therefore, in the current study, we assumed a homogeneous elastic and isotropic material for the degeneration disc [23,24].

The implanted FE model simulated the case in which the HPSS was used to treat a degenerated spine section in which the L3/L4 and L4/L5 discs were considered degenerated. As such, dynamic fixation was at L3/L4, whereas static fixation was at L4/L5. Tetrahedral meshing was performed for all screw–rod systems. The dynamic coupler was modelled as the flexible spring. The element size was chosen based on findings from a numerical convergence study.

In the present study, the influence of two variables on the biomechanical response of the model was determined. The first variable was the simulated state of degeneration of the L3/L4 and L4/L5 discs, which was reflected in the modulus of elasticity of the disc (E_m_), with E_m_ = 42, 32, and 20 MPa set for healthy, moderately degenerated, and severely degenerated discs, respectively. The second variable was K, with the values used being 28, 57, and 85 N/mm. The values of the modulus of elasticity (E) and Poisson’s ratio (ν) for each of the tissues in the FE models are presented in Table 1 [22,25,26]. 

Since the dynamic coupler stiffness in the HPSS represents an important parameter that affects the biomechanical characteristics of the implanted and adjacent segments after spinal non-fusion surgeries, it is necessary to evaluate the actual axial stiffness of the coupler as the input data for the analytical model. In this study, a universal materials testing machine (Electro Pulse E3000^®^; Instron, Inc., Norwood, MA, USA) was used to conduct the loading experiment and determine K. Figure 3 shows the experimental setup and the axial load-displacement curve of the coupler. The coupler presented a stiffness of 57 N/mm, which was determined as the slope of the initial portion of the load-displacement curve. This value was close to the original design dynamic coupler stiffness (50 N/m), which was developed based on a validated FEA model and verified by biomechanical experiments [18]. Therefore, in the current study, three different coupler stiffnesses for the HPSS—K_1_ = 28 N/mm, K_2_ = 57 N/mm, and K_3_ = 85 N/mm—were set as the parameters of the FE model to evaluate how the different coupler stiffnesses influenced the stress on the intrinsic degenerated disc.

### 2.2. Boundary and Loading Conditions

In our FE model, the inferior surface of the sacrum was rigidly fixed in all directions. The interfaces of the screw rod and the screw vertebra were designated as fully constrained. The superior surface of the T12 segment was set as a free end, and a loading that comprised flexion, extension, or lateral bending–loading and an axial compressive force of 500 N was applied to that surface, simulating a physiological loading [20,21]. The FE model was tested for mesh convergence. To obtain reliable results, the convergent analysis was conducted with the mesh model (T12–S1). A convergence analysis demonstrated that when the FE model had 919,139 elements with an element size of 1.8 mm, the maximum displacement approached the convergence value (error < 1%) and the mesh was convergent [27,28].

### 2.3. Validation of the FE Model

To validate the FE model, the human cadaveric spine (T12–S1) was subjected to 7.5 N-m moment (flexion, extension, and lateral bending) and an experimental investigation was performed into the load-displacement response. The uniaxial mechanical testing system (Instron, Electro Pulse E3000, Norwood, MA, USA) was used to conduct the applied loads. To produce a moment vector from the uniaxial tester, the stroke of the tester actuator, a cable–pulley mechanism was used to perform the pure moment in the test frame. A compressive follower load (FL = 500 N) was applied from T12–S1 using a system of eyelets, cables, pulleys, and dead weights to simulate body weight. The experimental setup and procedures were presented in our previous study [29,30]. In our FE model, the results of the range of motion (ROM) of the T12 to L5 vertebral body were predicted by a 7.5 N-m bending moment for flexion, extension, and lateral bending and combined with a 500 N axial load. The range of motion for each vertebral body was evaluated and compared with the previous experimental results. If the predicted results were within the standard deviation of the experiments, the FE model was validated. The results of the validation are presented in Figure 4. Thus, based on the comparison of the FE model with these experiments, the analytical results were similar to the data determined by the experiments. Therefore, the FE models developed in this study can be considered sufficiently accurate.

## 3. Results

### 3.1. Influence of Simulated State of Degeneration of L3/L4 and L4/L5 IVDs on von Mises Stress in IVDs

Figure 5 summarises the simulated results of degeneration at L3/L4 and L4/L5 IVDs. Three different dynamic coupler stiffnesses, 28 N/mm, 57 N/mm, and 85 N/mm, were individually installed at the L3/L4 level and fused at L4/L5. The von Mises stress in the intervertebral disc was evaluated under flexion and extension, and a lateral bending moment of 7.5 N/m was applied for the conditions of the disc with moderate degeneration (modulus of elasticity E_m_) and severe degeneration (modulus of elasticity E_s_). It was found that, in the HPSS with the coupler stiffness K_1_, all the intervertebral discs in the severely degenerated condition experienced higher stress compared with the moderately degenerated discs. However, there was a significant increase in von Mises stress at the adjacent level L2/L3. Similar results were observed in the system installed using the coupler stiffness of K_2_ and K_3_.

### 3.2. Influence of Simulated State of Degeneration of L3/L4 and L4/L5 IVDs on the Intersegmental Range of Motion (ROM) 

Figure 6 depicts the relationship between the mobility (range of motion) of each vertebral segment under flexion, extension bending, and lateral bending using the HPSS under two scenarios: moderate disc degeneration and severe disc degeneration, and couplers with three different stiffnesses were installed. Figure 6 demonstrates that, with the use of three different coupler stiffness settings, the mobility of the L4/L5 segment was relatively lower compared with that of the other vertebral discs. The mobility of the L3/L4 and L5/S1 was significantly increased under both moderate and severe degeneration conditions. The difference in ROM due to the different degradation grades seems to be insignificant.

### 3.3. Influence of Axial Stiffness of Dynamic Coupler (K) on von Mises Stress in IVDs

The coupler stiffness could have an impact on the mobility of intervertebral discs with different degeneration grades. Figure 7 shows the von Mises stress of the discs for the HPSS using three different dynamic coupler stiffnesses to investigate the optimal design of the coupler for the degenerative discs. The results show that, for the moderate disc, in the system with the coupler stiffness of K_2_, the adjacent intervertebral discs L2/L3 and L5/S1 had lower von Mises stress than those systems with coupler stiffnesses of K_1_ and K_3_. The severely degenerated condition showed a similar trend. However, the values of von Mises stress in the moderator degenerative disc were slightly lower than the values of severe degeneration.

### 3.4. Influence of Axial Stiffness of Dynamic Coupler (K) on the Intersegmental Range of Motion (ROM)

Figure 8 presents the results of the mobility of the vertebral segments under flexion, extension, and lateral bending, with three different coupler stiffness designs. The results show, that for the moderate disc, when the coupler stiffness was set to K_1_, a larger segmental range of motion could be maintained than that of the system with coupler stiffnesses of K_2_ and K_3_. The severely degenerated condition showed a similar trend.

## 4. Discussion

Adjacent segment diseases (ASDs) are prevalent after lumbar fusion surgery [1,2,4,7,10]. Dynamic spinal stabilisation systems have been developed to reduce the incidence of arthrodesis-related morbidity [12,13,14,15,16]. However, the efficacy of dynamic systems in the prevention of adjacent-level degeneration has not yet been proven. Several reports have established that spinal fusion with pedicle fixation accelerates the degeneration of adjacent motion segments because the relative immobility of fused spinal segments transfers stress to adjacent segments [2,9,10,11]. This study investigates the biomechanical effect of disc degeneration at the instrumented segments on the stress and range of motion of the adjacent segments fixed by the hybrid performance stabilisation system (HPSS) using finite element models. 

Figure 5 presents the results of the von Mises stresses of the vertebral segments subjected to a 7.5 N-m flexion, extension, and lateral bending for discs with moderate (E_m_ = 32 MPa) and severe degeneration (E_s_ = 20 MPa) using three different coupler stiffness designs. There was a significant increase in stress at the L5/S1 level under both moderate and severe degeneration. Figure 5 also shows that, under extension and lateral bending, the differences in disc stress caused by the different coupler stiffnesses were not significant. Based on these results, it can be preliminarily inferred that when an intervertebral disc has degeneration, under a flexion motion, using the three design stiffnesses will result in lower disc stress; however, the variation in stress with changing coupler stiffness was not significant under extension and lateral bending. In the adjacent level (L2/L3), the von Mises stress in the severely degenerated discs was higher than that of the moderately degenerated discs. The disc stresses under severe degeneration were approximately 25%, 33% and 30% higher than those under moderate degeneration corresponding to the coupler stiffness of K_1_, K_2_, and K_3_. However, in the fused segment (L4/L5), the stress on the disc was approximately 1 MPa, indicating that the stress shielding effect was not as pronounced as that for the moderate degeneration. It can be recommended that, in patients with severe disc degeneration using the dynamic spine stabilisation system, the intervertebral disc in the fusion segment still experiences some load, thereby revealing the stress-sharing phenomenon. This result suggests that the spinal dynamic stabilisation system is more suitable for patients with a moderate level of disc degeneration.

Figure 6 presents the results of the mobility of the vertebral segments under flexion, extension, and lateral bending for moderator and severely degenerated discs with three different coupler stiffness designs. In Figure 6, during flexion, the mobility of the adjacent segment (L2/L3) for moderate degeneration was higher than that in the severely degenerated disc. With coupler stiffnesses of 28, 57, and 85 N/mm, the mobility increased by 1%, 21%, and 29%, respectively. In the extension motion model, the mobility of the adjacent segment (L2/L3) also increased by 2%, 36%, and 8% with the respective coupler stiffness settings. However, there was no significant trend in the increased mobility observed in lateral bending. In this analysis, the mobility of the adjacent segment (L3/L4) also decreased as the coupler stiffness increased. For the extension and lateral bending, there was no significant difference in the mobility of the vertebral segment among the three different coupler stiffness settings. The higher coupler stiffness may decrease the range of motion of the severe degenerative disc at the adjacent level. 

In the moderate disc degeneration case, when the coupler stiffness (K_2_) was used in the HPSS, the intervertebral disc stress in the adjacent segment (L2/L3) was lower than those using coupler stiffnesses of K_1_ and K_3_ (Figure 7). Similarly, in the severe degeneration case, the analytical results showed a similar trend. This result suggests that, even when the spinal dynamic stabilisation system was designed with a coupler stiffness K_2_, it was more effective in reducing intervertebral disc stress in patients with disc degeneration. The coupling device K_2_ used in this study represents the “Paradigm Hoffman–PNS stabilisation system” currently used in clinical practices. The results indicate that, from the perspective of reducing intervertebral disc stress in the adjacent segment, patients with moderate disc degeneration are more suitable for the current HPS dynamic spinal stabilisation system compared with those with severe degeneration. When the coupler stiffness was changed to K_3_, the intervertebral disc stress in the adjacent segment (L2/L3) became higher in cases with severe disc degeneration compared with moderate degeneration during flexion, extension, and lateral bending. This result emphasises that, from the perspective of increased stress, the use of the dynamic spinal stabilisation system is more suitable in patients with moderate disc degeneration.

Figure 8 demonstrates the moderator degeneration, where the mobility of the L3/L4 level decreased by 27% when the coupler stiffness of the HPS system was set to 85 N/mm compared with 27 N/mm. Similarly, there was a noticeable difference in the T12/L1 level, with an overall mobility reduction of 14%. The same trend was observed in the results for extension movement, with the mobility of the L3/L4 and T12/L1 decreasing by 31% and 16%, respectively. Furthermore, in Figure 8, the results of the lateral bending show that the mobility for all the vertebral segments was significantly lower compared with the flexion and extension. This is because the facet joints in the lumbar vertebrae are more upright compared with the cervical and thoracic spine, which limits lateral mobility. The results also indicate that the coupler in the HPS system restricts lateral spinal bending.

Based on these findings, it can be summarised that, regardless of whether the coupler stiffness is set to K_1_, K_2_, or K_3_, in cases of severe disc degeneration, the disc stress in each spinal segment will be higher compared with cases of moderate disc degeneration. The result suggests that the current clinical HPS system is suitable for use in patients with moderate disc degeneration. However, those patients with moderate disc degeneration who do use the HPS system may have lower mobility in adjacent segments compared with patients with severe degeneration. A higher stiffness in the coupler in the HPSS was shown to reduce the mobility of the non-fused segments, thereby affecting overall mobility. This study found that mobility performance does not directly affect the stress concentration in the adjacent intervertebral discs (L2/L3). However, it may have an impact on daily activities.

Demir et al. [21] used a finite element (FE) model to predict the variations in ROM and intradiscal pressure (IDP) from intact to implanted situations when using a hybrid stabilisation system (HSS) for the human lumbar spine. In their implanted model, the L4–L5 level was implanted dynamically and the L5–S1 level was subjected to fusion and fixed with rigid stabilisation. They concluded that, after HSS implantation, the L4–L5 level did not lose its motion completely, while L5–S1 had no mobility as a consequence of the disc removal and fusion process. Although our study used the HPSS system, our study showed similar results in that the dynamic level (L3/L4) retained the ROM and limited mobility of the fusion level (L4/L5). 

Experimentally, it is difficult to measure the stress within the vertebral body. However, the stress in the intervertebral disc can be measured using the compressive force through the endplate of the vertebral body [29]. When it comes to compression loading in the spine motion segment, age-related degeneration may lead to a transfer of load from the nucleus to the posterior annulus [23,31,32]. Nonetheless, experiments conducted on vertebrae with both healthy and degenerated discs have shown that disc stress does not vary significantly across the entire disc [33,34,35]. Moreover, the degree of disc degeneration does not affect the load distribution between the anterior and posterior halves of the disc [36,37]. Landham et al. suggested that, in the case of physiological flexion loading, the differences between healthy and degenerated intervertebral discs appeared to have little impact on the overall stress on the vertebral endplates or within the vertebral body [38,39,40]. Thus, the fine details of real intervertebral disc behaviour may not significantly influence overall stress. Studies have shown that the predicted distributions for disc stress and the locations of the initial bone failure are consistent with cadaver experiments subjected to a moderate flexion moment to both healthy and degenerated discs. These comprehensive findings indicate that, despite the relatively simple modelling approach for the intervertebral disc, our results can reasonably be applied to understand the impact of both healthy and degenerated discs on vertebral fractures.

Furthermore, in the case of axial rotation (AR), the posterior fixation may result in different rotation centres between the adjacent vertebral bodies and the fixed vertebral body, which are potentially accompanied by other displacements. Most posterior-based spinal fixation systems are primarily designed to improve flexion and extension movements. In the case of lateral bending (LB), asymmetric forces may lead to unexpected coupling effects when there is deformation or asymmetrical displacement of the implant. This can cause not only lateral curvature (or lateral rotation) but also phenomena such as rotation or flexion/extension (FL/EX). Additionally, under an axial rotation, the posterior fixation changes the original rotation centre of the spine, resulting in a difference between the adjacent vertebral bodies and the rotation centre of the vertebral body [41,42,43,44]. This difference may be accompanied by additional displacements of the vertebral bodies, amplifying the coupling effect during lateral bending or rotation [18]. Thus, there is an increased relative motion of adjacent facets or stress on the posterior rod and altered load distribution between the vertebral bodies and the implant. These coupled behaviours may reduce the functionality provided by the originally designed dynamic stabilisation system or promote unexpected biomechanical responses.

This study has several limitations. First, the FE model did not include muscles, and surgically induced changes in segmental and pelvis kinematics, as well as muscle cross-sectional areas (due to intraoperative iatrogenic injuries), were not considered in our model [9]. Second, only the axial stiffness of the coupler was varied. The flexural stiffness and its torsional stiffness were not considered. In the current model, a spring was employed to simulate the coupler of the HPSS, and a more sophisticated design was needed to limit unwanted effects on spinal functional segments. Third, the changes in the biomechanical responses determined concerned a model of a healthy spine section, rather than a model of a spine section that contained the two degenerated IVDs. Little is known regarding the change in the material properties of all discs and spinal ligaments due to degeneration. The FE model was only developed based on the assumptions and simplifications concerning the geometry, material properties, and applied loads. Fourth, quasi-static mechanical testing was performed on the coupler, rather than dynamic tests, such as the fatigue life and fatigue crack propagation rate. Fifth, the loading conditions in our model did not precisely match the constraint conditions in the actual living bodies. In reality, the direction and magnitude of the lumbar muscle forces were constantly changing and responding to the stability requirements of the spine. Although the path of the follower load was applied along the curvature of the spine, the static (constant) follower load used in our test was under a simulated biomechanical condition, not under a physiological condition. Lastly, in our model, a single fused level was created at L4/5; however, different fusion lengths could affect the range of motion and disc stress at the adjacent levels. A multi-level fusion model will help to clarify the further effect of the posterior-based hybrid performance stabilisation system.

## 5. Conclusions

(1)For discs with moderate degeneration, increasing the coupler stiffness led to a decrease in disc mobility. However, in the case of severe disc degeneration, the effect on disc mobility by coupler stiffness was less pronounced. While considering the design and selection of coupler stiffness, the degeneration level of intervertebral discs should be considered to achieve the desired mobility characteristics;(2)Increasing the coupler stiffness led to higher stress on intervertebral discs with moderate degeneration, while its effect on stress was less pronounced for discs with severe degeneration. In flexion, extension, and lateral bending, higher coupler stiffness resulted in a reduction in the mobility of adjacent segments.

Further experimental analysis and clinical research are needed to confirm the biomechanical characteristics associated with using the hybrid spinal stabilisation system.

## Figures and Tables

**Figure 1 bioengineering-10-01042-f001:**
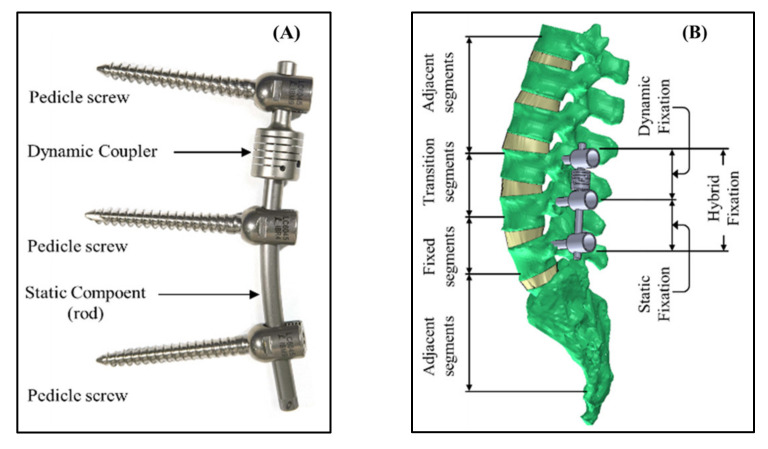
(**A**) The dynamic coupler (diameter = 14 mm; length = 14 mm; bar diameter = 5.4 mm; length = 80 mm) and static component of the hybrid performance stabilisation system (HPSS). (**B**) The implanted spinal model with hybrid (static and dynamic) fixation.

**Figure 2 bioengineering-10-01042-f002:**
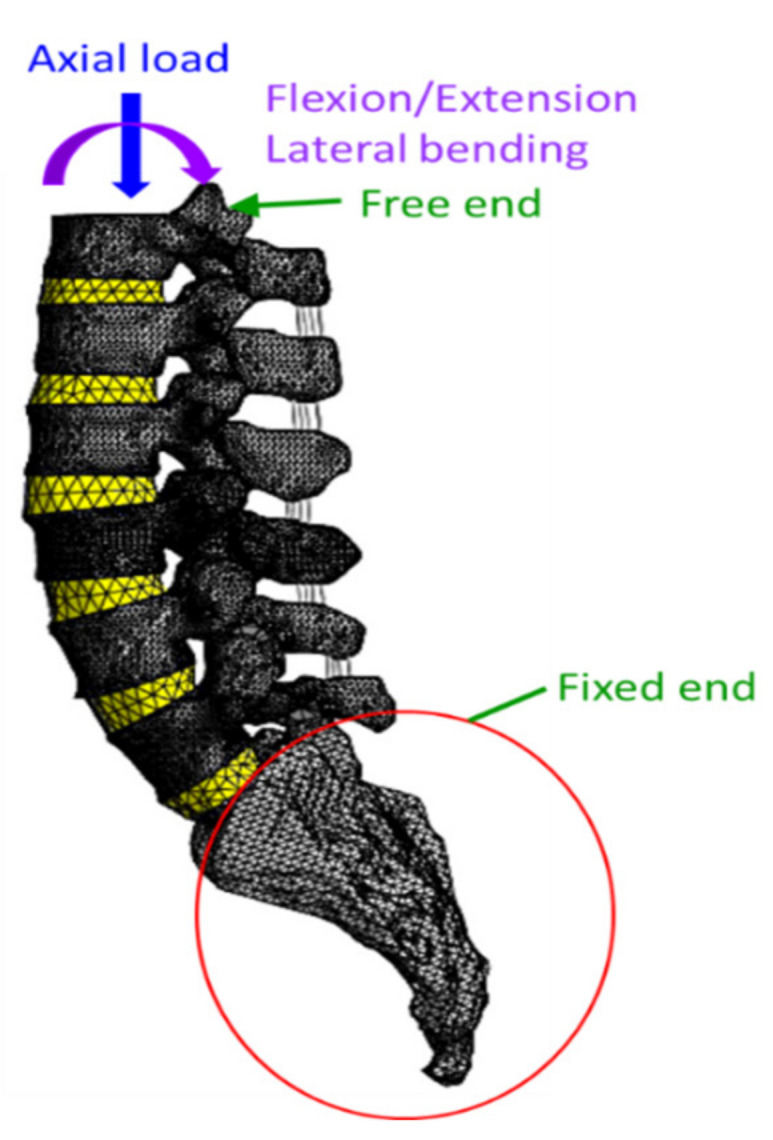
The meshed FE model and the boundary conditions.

**Figure 3 bioengineering-10-01042-f003:**
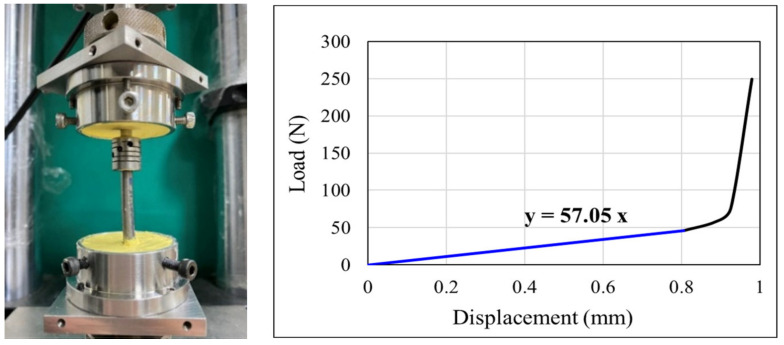
Close-up photograph of the dynamic coupler under axial load testing and a typical load-displacement curve obtained during a test.

**Figure 4 bioengineering-10-01042-f004:**
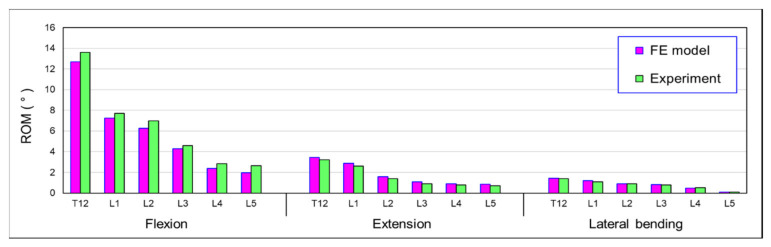
Summary of the model validation results.

**Figure 5 bioengineering-10-01042-f005:**
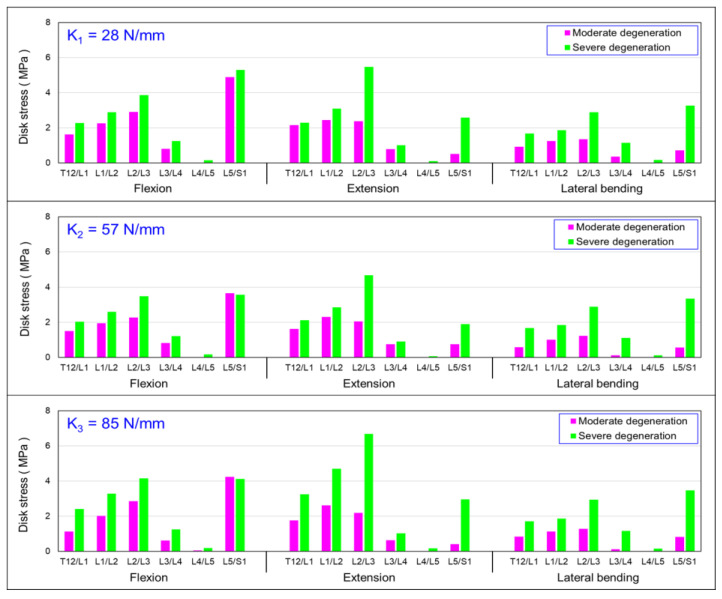
Summary of mean von Mises stress in the discs: influence of moderate and severe degenerated discs.

**Figure 6 bioengineering-10-01042-f006:**
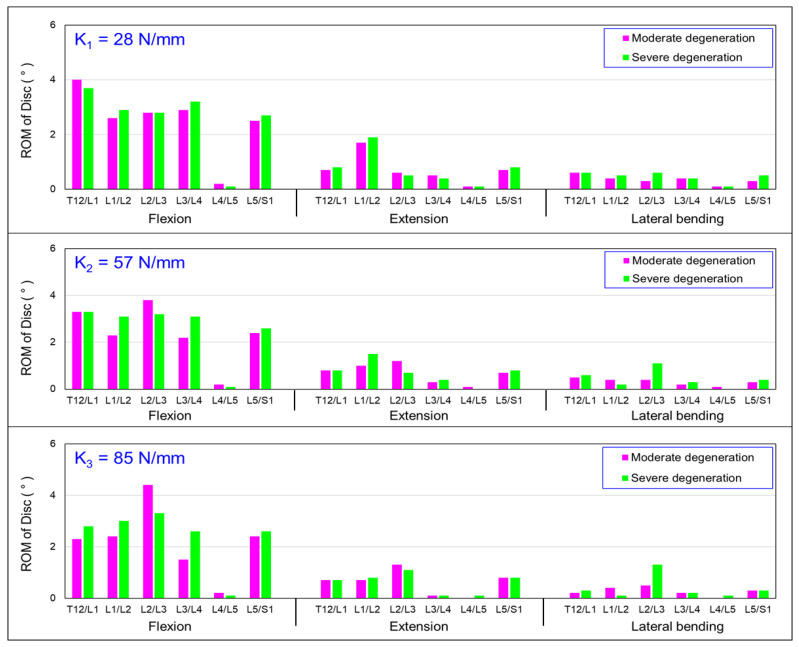
Summary of the change in the intersegmental range of motion (ROM) values in the discs: influence of moderate and severe degeneration.

**Figure 7 bioengineering-10-01042-f007:**
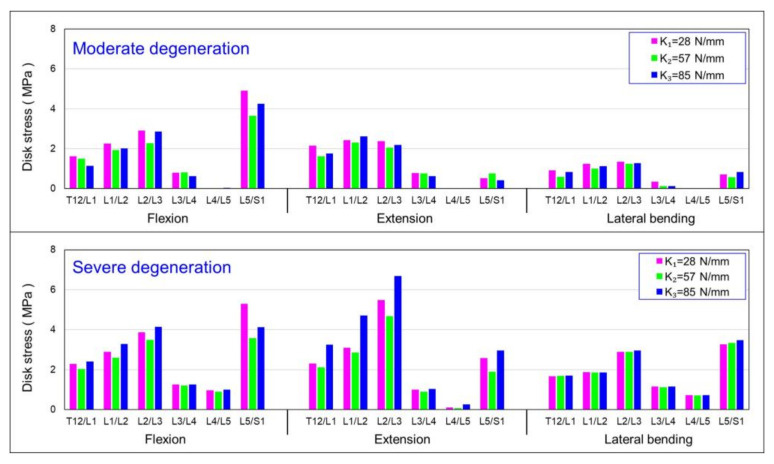
Summary of the change in mean von Mises stress values in the discs: influence of the stiffness of the dynamic coupler (K).

**Figure 8 bioengineering-10-01042-f008:**
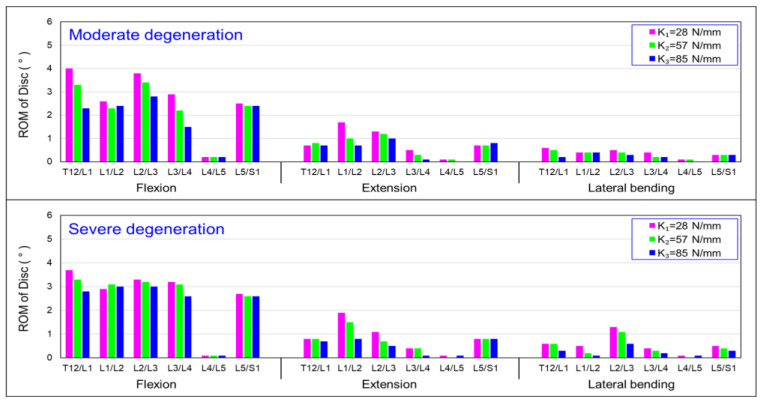
Summary of the change in the intersegmental range of motion (ROM) values in the discs: influence of the stiffness of the dynamic coupler (K).

**Table 1 bioengineering-10-01042-t001:** Material properties used in the FE model [22,25,26].

Material (Tissues)	Elastic Modulus (MPa)	Poisson’s Ratio	Element Number	Node Number
Cortical bone [25]	12,000	0.3	31,562	15,632
Cancellous bone [25]	100	0.2	28,033	13,355
Healthy annulus [25]	8.4	0.45	20,238	16,856
Nucleus pulposus [25]	1	0.499	--	--
Endplate [25]	24	0.4	6035	4366
Moderate degenerated disc (E_m_) [22]	32	0.3	266,322	166,451
Severely degenerated disc (E_s_) [22]	20	0.3	266,322	166,451
Ligament [26]	8	0.3	8963	5647
Screw, rod, and coupler [26]	110,000	0.28	40,569	25,396

## Data Availability

Data are contained within the article.

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
