# Peer review of "Influence of Simulated State of Disc Degeneration and Axial Stiffness of Coupler in a Hybrid Performance Stabilisation System on the Biomechanics of a Spine Segment Model"

_bioengineering, 2023, doi:10.3390/bioengineering10091042_

Round 1

Reviewer 1 Report (Previous Reviewer 2)

I have checked it . It is fine.

Author Response

Thanks for the reviewer's vital comments and for checking our revision.

Reviewer 2 Report (Previous Reviewer 3)

All issues satisfactorily addressed. Excellent Revised Manuscript.

However, a few minor editorial corrections are needed. These are listed in the attached file, "LIST-OF-MINOR-EDITORIAL-CORRECTIONS.DOCX".

Author Response

100 “…………in the HPSS (K) on changes in…..” has been revised.  (Line 101) 

162-163 “determine K. Figure 3 shows…………….” has been revised. (Line 162)

201 “…..are listed in Figure 4. Thus,………..” has been revised.  (Line 200)

227-228 “………using the HPSS in two scenarios:………………..” has been revised. (Line 227)
240 “the HPSS using three………………” has been revised.  (Line 239)

308 “in the HPSS, the intervertebral………….” has been revised.  (Line 307)

323 “……..demonstrates the moderate degeneration……..” has been revised.  (Line 322)

341 “HPSS was shown to reduce……………..” has been revised.  (Line 340)

347 “a hybrid stabilization system (HSS) for the…………….” has been revised.  (Line 346)

351-352 “………….used the HPSS, however, our………………” has been revised.  (Line 340-351)

This manuscript is a resubmission of an earlier submission. The following is a list of the peer review reports and author responses from that submission.

Round 1

Reviewer 1 Report

This article is interesting and well written. In particular, the procedures that led to the creation of the HPS system are well explained in detail. However, further information regarding the size of the sample should be provided.

Tables in figure 6 are overlaid, preventing the correct view of the data

Author Response

Reviewer #1

Comments and Suggestions for Authors

This article is interesting and well-written. In particular, the procedures that led to the creation of the HPS system are well explained in detail. However, further information regarding the size of the sample should be provided.

Reply: Thanks for the reviewer's comments. In this study, one human spine cadaveric specimen was used for validation of the FE model. The size of the dynamic coupler has been added to the revised manuscript. (Line 122-123)

Tables in Figure 6 are overlaid, preventing the correct view of the data

Reply: The overlay in Fig. 6 has been revised.

Reviewer 2 Report

The paper entitled, ‘Biomechanical Effect of Coupler Stiffness of Hybrid Performance Stabilization (HPS) System on The Intrinsic Degenerated Adjacent Segments in Lumbar Spine’ aimed to investigate the biomechanical effect of the coupler stiffness of the HPS system on the mobility of the lumbar vertebral segments and the stress distribution in the intervertebral discs under flexion, extension, and lateral bending, as well as the clinical applicability of the HPS system on the discs with intrinsic moderate and severe degeneration at the adjacent level. The manuscript has been written nicely. I have some points which may make this piece of research more readable with clarity.

1.         Authors should mention the reason for accelerated disc degeneration and adjacent segment fractures due to fusion surgery.

2.         Introduction section lacks proper citations, however, authors have cited references collectively in between, but I suggest that instead of giving references such as 1-5, 13-18, and 19-25, authors must cite individual references at the end of the sentence.

3.         Authors must cite reference after line ……Although the coupler in the adjacent segment (line 87)

4.         In Figure 3, comparisons of FE predicting and experimental results, legend should use the word, prediction rather than predicting. I suggest that these three figures can be combined into one figure by taking vertebrae on X-axis and degrees on Y-axis and adjusting bar diagrams of all three; flexion, extension, and lateral bending with different colors.

5.         Authors must include a Discussion section in the manuscript, which can be started from line 304 onwards.

6.         Authors must add the scope and limitations of the study. This would clarify the extent to which the findings can be generalized and any potential constraints in the research methodology.

Author Response

Response to Reviewer 2 Comments

Reviewer #2

Comments and Suggestions for Authors

The paper entitled, ‘Biomechanical Effect of Coupler Stiffness of Hybrid Performance Stabilization (HPS) System on the Intrinsic Degenerated Adjacent Segments in Lumbar Spine’ aimed to investigate the biomechanical effect of the coupler stiffness of the HPS system on the mobility of the lumbar vertebral segments and the stress distribution in the intervertebral discs under flexion, extension, and lateral bending, as well as the clinical applicability of the HPS system on the discs wif intrinsic moderate and severe degeneration at the adjacent level. The manuscript has been written nicely. I have some points which may make dis piece of research more readable with clarity.

  1. The authors should mention the reason for accelerated disc degeneration and adjacent segment fractures due to fusion surgery.

Reply: Thanks for the reviewer's valuable comments.

Based on the clinical finding and biomechanical studies, this phenomenon could be due to the stresses raising or called stress concentration at the adjacent level of the fusion region. Some studies have found that adjacent segment diseases may be caused by spinal fusion, which can induce abnormal intradiscal pressure and too much movement at the adjacent spinal levels, resulting in abnormal discal stress distribution. This is because the fusion segments form a rigid region, restricting the range of motion at the fusion levels, however, the adjacent segments have to compensate for the lost motion of the fusion levels. Then the higher stress and larger motion were excited at the adjacent level. The reasons for accelerated disc degeneration and adjacent segment fractures due to fusion surgery have been added in the “introduction”. (Lines 46-54)

  1. The introduction section lacks proper citations, however, authors have cited references collectively in between, but I suggest that instead of giving references such as 1-5, 13-18, and 19-25, authors must cite individual references at the end of the sentence.

Reply: We have revised the citation as individual references at the end of the sentence. The citation style was according to the author’s instructions in “Bioengineering” Journal.

  1. Authors must cite reference after line ……Although the coupler in the adjacent segment (line 87)

Reply: The reference [19] has been cited at the end of the sentence.

  1. In Figure 3, comparisons of FE predicting and experimental results, the legend should use the word, prediction rather than predicting. I suggest that these three figures can be combined into one figure by taking vertebrae on X-axis and degrees on Y-axis and adjusting bar diagrams of all three; flexion, extension, and lateral bending with different colours.

Reply: Thanks for the reviewer’s suggestion, Figure 3 has been revised as the suggestion.

  1. Authors must include a Discussion section in the manuscript, which can be started from line 304 onwards.

Reply: The “Discussion” section has been added in the manuscript. (Line 305)

  1. Authors must add the scope and limitations of the study. This would clarify the extent to which the findings can be generalized and any potential constraints in the research methodology.

Reply: The limitation of this study has been added in the manuscript. (Line 352-369)

Reviewer 3 Report

PLEASE SEE ATTACHED WORD DOCUMENT, "REVIEW-BIOENGINEERING-MANUSCRIPT-#-2505958.DOCX".

Quality is good but several editorial revisions and corrections are needed.